

# Advances in cultivation, wastewater treatment application, bioactive components of *Caulerpa lentillifera* and their biotechnological applications

Xiaolin Chen[1,2,3], Yuhao Sun[1,2,3,4], Hong Liu[1,2,3,4], Song Liu[1,2,3], Yukun Qin[1,2,3] and Pengcheng Li[1,2,3]

[1] CAS Key Laboratory of Experimental Marine Biology, Institute of Oceanology, Chinese Academy of Sciences, Qingdao, China
[2] Laboratory for Marine Drugs and Bioproducts of Qingdao National Laboratory for Marine Science and Technology, Qingdao, China
[3] Center for Ocean Mega-Science, Chinese Academy of Sciences, Qingdao, China
[4] University of Chinese Academy of Sciences, Qingdao, China

Corresponding authors
Xiaolin Chen, chenxl@qdio.ac.cn
Pengcheng Li, pcli@qdio.ac.cn

## ABSTRACT

The edible seaweed *Caulerpa lentillifera*, a powerful natural food source that is rich in protein, minerals, dietary fibers, vitamins, saturated fatty acids and unsaturated fatty acids, has been mass cultured in some Asian countries and has been the focus of researchers in recent years. Here, the operational conditions of its culture, application in wastewater treatment, and bioactive components are summarized and comparatively analyzed. Based on previous studies, salinity, nutrient concentrations, irradiance and temperature are stress factors for algal growth. Moreover, dried *Caulerpa lentillifera* seaweed is efficient in the biosorption of heavy metals and cationic dyes in wastewater, and fresh seaweed can be introduced as a biofilter in aquaculture system treatment. In addition, among the rich bioactive compounds in *Caulerpa lentillifera*, the phenolic compounds show the potential ability for regulating glucose metabolism in vivo. Polysaccharides and oligosaccharides exhibit anticoagulant, immunomodulatory effects and cancer-preventing activity. Siphonaxanthin is a compound with attractive novel functions in cancer-preventing activity and lipogenesis-inhibiting effects. Furthermore, the antioxidant activity of siphonaxanthin extracted from *Caulerpa lentillifera* could be stronger than that of astaxanthin. This review offers an overview of studies of *Caulerpa lentillifera* addressing various aspects including cultivation, wastewater treatment and biological active components which may provide valuable information for the cultivation and utilization of this green alga.

## INTRODUCTION

As shown in Fig. 1, *Caulerpa lentillifera*, green seaweed with high economic value, is naturally distributed in tropical and subtropical regions, such as South China Sea, Southeast Asia, Japan, Okinawa, Taiwan and Oceania (*Paul et al., 2014*). As reported in literatures, this green seaweed was documented for the first time on Red Sea coast (*Agardh, 1837*), and

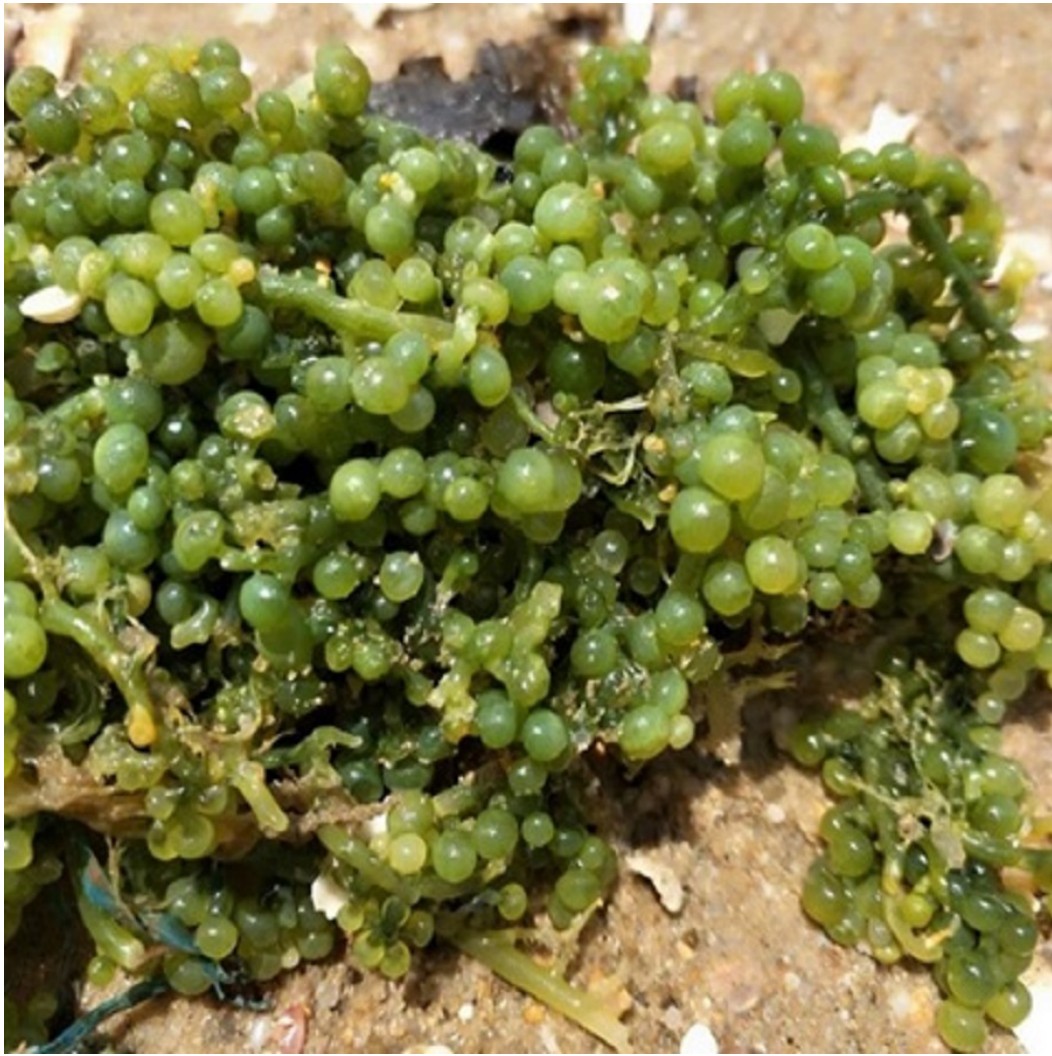

**Figure 1** *Caulerpa lentillifera* grown in Huang Hai, China (supplied by Xiaolin Chen).

then was observed at many other locations, especially in Indo-Pacific region (*Hackett, 1977*; *Taylor, 1977*; *Menez & Calumpong, 1982*; *Coppejans & Beeckman, 1990*; *Phillips, Conacher & Horrocks, 1999*; *Schils & Coppejans, 2003*; *Titlyanov, Titlyanova & Pharm, 2012*). Because its upright branches resemble grapes, *C. lentillifera* is also called "sea grapes" (*Guo et al., 2015a*), and it can grow on sand and rock bottoms in the upper sublittoral zone of tropical coral reefs (*Horstmann, 1983*; *Mao et al., 2011*). Because of its good taste, *C. lentillifera* is often cooked as salad in some Asian countries. In addition, *C. lentillifera* is rich in polyunsaturated fatty acids (PUFAs) (*Saito et al., 2010*), multiple essential amino acids, minerals, dietary fibers, vitamin A and Vitamin C (*Matanjun et al., 2009*) and has low levels of lipids (*Niwano et al., 2009*). Therefore, there has been increasing demand and rising market prices for *C. lentillifera* in some Asian countries recently. However,

although this alga is widely cultivated in Philippines (*Zemkewhite & Ohno, 1999*), Okinawa (*Kurashima et al., 2003*), Taiwan Island (*Shi, 2008*), Fujian and Hainan provinces in China (*Wang, 2011*), the commercial-scale production of *C. lentillifera* is still not sufficient, and its productivity does not meet the demand. It might be due to lack of optimum cultivation conditions of the alga. Therefore, it is important to obtain the best culture conditions to increase the productivity of *C. lentillifera*.

Currently, the main research of this alga focuses on the treatment of wastewater and development of bioactive components. *C. lentillifera* has shown potential ability to remove basic dyes from waste streams (*Marungrueng & Pavasant, 2006*), heavy metals from industrial wastewater (*Pavasant et al., 2006*; *Apiratikul & Pavasant, 2008*), and nutrients from aquaculture effluents (*Paul & De Nys, 2008*), especially $NO_3$-N (*Guo et al., 2015b*).

In recent years, some bioactive components of *C. lentillifera*, such as phenolic compounds, polysaccharides and pigments, and their biological potentials, including antioxidant, anti-diabetic and anticancer activities, have been documented. Therefore, in order to further understand and make better use of this seaweed, we summarized researches of its cultivation conditions, wastewater treatment abilities, and bioactive components along with their biological activity.

## METHODS

### Survey methodology

Two main databases were used to obtain related literature including Web of Science and Google Scholar from 1900–2018. The selected references were listed after the acknowledgement. From the previous researches, the results were reviewed.

### Cultivation conditions for *C. lentillifera*

According to previous literatures (shown in Table 1), we concluded that salinity, nutrient concentration, irradiance and temperature were all stress factors for growth during all periods, when these factors change and they will correspondingly affect the physiology of the alga such as growth rate, chlorophyll concentration etc. Therefore, it is important to study the optimum factors for massive culture of the alga. *Deraxbudsarakom et al. (2003)* suggested that a salinity range of 25–30‰ was suitable for the normal growth of *C. lentillifera* when the alga was cultured by shrimp farm effluent at laboratory. *Wang (2011)* showed that the maximum growth of *C. lentillifera* supplied by Fujian China occured at a salinity of approximately 36‰, which was cultured with filtered seawater added with salt; later, a study by *Guo et al. (2015b)* confirmed this result. C. *lentillifera* transported from Okinawa Japan did not survive at salinities of 5‰ and 55‰ cultured by sterile seawater. This study indicated that the specific growth rate (SGR) for *C. lentillifera* was different among all the groups. The maximum SGR was obtained at a salinity of 35‰, and this result was consistent with the maximum chlorophyll content and the ratio of fluorescence (Fv/Fm). At salinities of 20‰ and 45‰, only stolons regenerated from branches. However, new branches grew from stolons at salinities 30‰–40‰. Therefore, studies suggested that the optimal salinity concentration for the growth of *C. lentillifera* was between 30‰–40‰.
**Table 1   The effect of cultivation conditions on *C. lentillifera*.**

| Cultivation conditions | Effect | Reference |
|---|---|---|
| Salinity | Suitable salinity range of 25–30‰; the maximum growth at a salinity of approximately 35–36‰ | *Deraxbudsarakom et al. (2003)*, *Wang (2011)* and *Guo et al. (2015b)* |
| Nitrogen and phosphorus | Optimal for the rapid growth at 0.6 mmol/L $NO_3$-N and N:P ratio of 8:1; Highest SGR at a 0.1 mmol/L $PO_4$-P and 0.5 mmol/L $NO_3$-N; Nitrogen types ($NaNO_3$ and $NH_4NO_3$) can significantly promote the growth of the alga; $NH_4$-N:$NO_3$-N ratios of 1:1 and 1:5 were the most favorable ratios for the growth of the alga | *Deraxbudsarakom et al. (2003)*, *Guo et al. (2015b)*, *Wang et al. (2017)*, *Liu et al. (2016)* |
| Phytohormones | 6-BA and GA could induce the growth of the alga, but IAA could increase the intracellular crude polysaccharide content | *Tao et al. (2017)* |
| Temperature | Ideal temperature range 22–28 °C | *Friedlander et al. (2006)* and *Guo et al. (2015a)* |

Nitrogen (N) and phosphorus (P) are two essential nutrients for the growth of *C. lentillifera* and they must be taken from the environment. *Deraxbudsarakom et al. (2003)* concluded that a 0.6 mmol/L $NO_3$-N concentration and N:P ratio of 8:1 were optimal for the rapid growth of *C. lentillifera* with salinity 25–30‰. However, *Guo et al. (2015b)* reported that SGR of *C. lentillifera* from Okinawa was the highest at 0.1 mmol/L $PO_4$-P concentration and 0.5 mmol/L $NO_3$-N concentration (approximate N:P ratio of 5:1, water temperature 25 °C and light of 40 umol photons/(cm$^2$ s)), which was slightly different from the results of *Deraxbudsarakom et al. (2003)*. In addition to nitrogen concentration, different oxidation states of N also had effects on the biomass production of *C. lentillifera*. For example, *Wang et al. (2017)* used four different nutrient salts ($NaNO_3$, $NH_4NO_3$, $CO(NH_2)_2$ and $NH_4HCO_3$) to cultivate *C. lentillifera* supplied by Ocean University of China with temperature 27 °C, light of 145.45 umol photons/(cm$^2$ s) and salinity 30‰. The results showed that nitrate ($NaNO_3$ and $NH_4NO_3$) can significantly promote the growth of the alga. Under a concentration of 20 mg/L $NH_4NO_3$, the relative growth rate of the alga was the highest. In addition, *Liu et al. (2016)* indicated that $NH_4$-N: $NO_3$-N ratios of 1:1 and 1:5 were the most favorable ratios for the growth of the alga. In conclusion, the optimal concentration of $NO_3$-N was 0.1 mmol/L-0.6 mmol/L and the optimal N: P was 5:1-8:1 for the growth of *C. lentillifera*.

Different phytohormones, such as gibberellin (GA), 6-benzyl aminopurine (6-BA) and indoleacetic acid (IAA), have also been shown to be efficient for the growth of *C. lentillifera* (*Tao et al., 2017*). The results (*Tao et al., 2017*) revealed that 0.8 and 1.4 mg/L 6-BA could induce a relatively high weight gain rate and SGR of *C. lentillifera* and that 11 mg/L GA was the optimal concentration for rapid growth, while IAA showed no obvious effect on the biomass of *C. lentillifera*. In addition, compared to GA, which had no significant effect on the production of crude polysaccharides in *C. lentillifera*, IAA increased the intracellular crude polysaccharide content.

Temperature has a major effect on the kinetics of cellular enzymes, and irradiance is an essential source of photosynthetic activity in algae. Hence, the growth of *C. lentillifera* is also induced by temperatures and irradiances at certain degrees. A previous study

**Table 2  Different wastewater treatment process by *C. lentillifera*.**

| Alga types | Wastewater types | References |
|---|---|---|
| Dried alga | $Cu^{2+}$, $Cd^{2+}$, $Pb^{2+}$ and $Zn^{2+}$ | *Pavasant et al. (2006)* and *Apiratikul & Pavasant (2006)* |
| Dried alga | Cationic dyes: Astrazon Blue FGRL (AB), Astrazon Red GTLN (AR), and methylene blue (MB) | *Marungrueng & Pavasant (2006)*, *Marungrueng & Pavasant (2007)*, *Ncibi, Mahjoub & Seffen (2007)*, *Cengiz & Cavas (2008)*, *Punjongharn, Meevasana & Pavasant (2008)* |
| Fresh alga | Used as a biofilter in aquaculture systems for nutrient absorption, especially $NO_3$-N | *Paul & De Nys (2008)*, *Liu et al. (2016)*, *Chokwiwattanawanit (2000)* |

showed that *C. lentillifera* started to become soft and decay and the productivity of biomass decreased sharply when the temperature reduced to 18 °C. Moreover, *Guo et al. (2015a)* found that the biomass of the alga reached the maximum of $6.932 \pm 0.396\%$ $day^{-1}$ at 27.5 °C and 40 μmol photons/(m² s). In addition, the authors also found that higher irradiances (40–100 umol photons/(m² s)) could decrease the chlorophyll content and *rbcL* expression. An experiment by *Wu et al. (2017)* further confirmed that different levels of light quality showed different effects on the growth and photosynthetic pigment contents of *C. lentillifera*. The concrete results showed that the light treatment of a blue/red ratio of 5/1 had significant beneficial effects on the fresh weight/length ratio, the fresh weight of regenerated vertical branches and the diameter of regenerated spherical ramuli. However, the contents of total chlorophyll, chlorophyll a, chlorophyll b and carotenoids significantly increased under full blue light. A comprehensive analysis suggested that 5/1 for blue/red and full white were suitable for indoor culture of *C. lentillifera*. In summary, the optimal temperature for the growth of *C. lentillifera* was about 20–28 °C. And more blue light or full white treatment would be benefit for the cultivation.

Besides the above cultivation parameters, the origin of the alga such as different area might lead to different growth results. However, there was no reference to introduce the research.

With the development of culture research, different applications of *C. lentillifera* have been studied. And wastewater treatment was early studied.

## Wastewater treatment by *C. lentillifera*

As mentioned in documents, *C. lentillifera* has been studied as biosorption material to treat wastewater, such as heavy metal wastewater, toxic dye-contaminated wastewater and aquaculture wastewater(shown in Table 2). There are several advantages to apply seaweeds as biosorbent, including their wide availability, low cost, high metal sorption capacity, reasonably regular quality, and relatively simple application. *Pavasant et al. (2006)* proved the ability of dried *C. lentillifera* to absorb $Cu^{2+}$, $Cd^{2+}$, $Pb^{2+}$ and $Zn^{2+}$. Moreover, the removal efficiency of the alga rose with an increased pH 2–8 (temperature $21 \pm 2$ °C), and the sorption process of all metal ions only took 20 min which was much faster than that of alginate/Mauritanian clay (with diffusion coefficient $4–8 \times 10^{-7}$ cm²/S; *Ely et al., 2011*). The sorption of heavy metals on the biosorbents mainly included two steps (*Pavasant et al., 2006*):

1. The metal ions were initially taken up onto the surface of the cells;

**Table 3  The values for $q_e$, $k$ and $R^2$ of Cu$^{2+}$, Cd$^{2+}$ and Pb$^{2+}$ in pseudo second-order kinetic mode.**

| Parameters | Cu$^{2+}$ | Cd$^{2+}$ | Pb$^{2+}$ |
|---|---|---|---|
| $q_e$ (mmol Kg$^{-1}$) | 6.14 | 3.97 | 2.64 |
| $K$ (Kg mol$^{-1}$ min$^{-1}$) | 254 | 621 | 2,036 |
| $R^2$ | 0.999 | 1.000 | 1.000 |

2.  They were bioaccumulated within the cells due to the metal uptake metabolism.

Step 1 involved passive transport, and it took place quite rapidly, i.e., within 20–30 min, while Step 2 took much longer to complete. In this case, the alga was dried and no longer active, so the sorption could only take place on the surface of the cell, which controlled the whole sorption process. Therefore, it took place only 20 min. Furthermore, the sorption process followed the Langmuir isotherm, and the maximum sorption capacities were Pb$^{2+}$ > Cu$^{2+}$ > Cd$^{2+}$ > Zn$^{2+}$.

In another study, the authors (*Apiratikul & Pavasant, 2008*) continued to use dried *C. lentillifera* to study the biosorption process of Cu$^{2+}$, Cd$^{2+}$ and Pb$^{2+}$, and the sorption kinetics best followed the pseudo second-order kinetic model:

$$q = \frac{q_e^2 kt}{1 + q_e kt} \tag{1}$$

In Eq. (1), $q$ (mg/g) is the amount of the metal adsorbed at time $t$ (min), $q_e$ (mmolKg$^{-1}$) is the amount of the metal adsorbed at the time of equilibrium, and k is the equilibrium rate constant. The values for $q_e$, $k$ and $R^2$ were listed in Table 3.

In addition, the sorption isotherm data fit the Langmuir isotherm model:

$$q_e = \frac{q_{max}Ce}{1 + bc_e} \tag{2}$$

In Eq. (2), $q_e$ represents the amount of metal ion taken up per unit mass of the biomass at equilibrium (mol/kg), $q_{max}$ is the maximum amount of metal ion taken up per unit mass of the biomass (mol/kg), b is the Langmuir affinity constant (m$^3$/mol), and Ce is the equilibrium concentration of the heavy metal ion in solution (mol/m$^3$). In addition, according to Dubinin-Radushkevich model, the sorption energies are 4–6 kJ/mol, as the process involves a physical electrostatic force. Ion exchange is believed to be a principal mechanism of the sorption, and metal ions such as Ca$^{2+}$, Mg$^{2+}$ and Mn$^{2+}$ are the main ions released from the algal biomass. In addition, the binary component systems composed of Cu$^{2+}$, Cd$^{2+}$ and Pb$^{2+}$ were also studied for the sorption of dried *C. lentillifera*. The experimental data was effectively described by the partial competitive binary isotherm model. In addition, the secondary metal ion always reduced the total sorption capacity of the previous metal ions, which implied that the concomitant metal ions competed for the same pooled binding sites during the algal biomass sorption process, and Pb$^{2+}$ was the most adsorbed metal ion according to the study. The batch scale experiments by fixed bed column also showed that sorption capacities for various metals could also be prioritized with the same order: Pb$^{2+}$ > Cu$^{2+}$ > Cd$^{2+}$. These results were beneficial for the further design and scaling up of the system (*Apiratikul & Pavasant, 2008*; *Apiratikul & Pavasant, 2006*).
Dried *C. lentillifera* has also been utilized to treat cationic dyes, which are widely used in the textile industry, because dried *C. lentillifera* contains many functional groups (O-H, COOH, NH$_2$ and S=O) that exhibit chemical binding affinity toward several positively charged ions, and these characteristics might also be showed by other algae. Overall, dried *C. lentillifera* was proved to effectively absorb Astrazon Blue FGRL (AB), Astrazon Red GTLN (AR), and methylene blue (MB). The maximum sorption capacity of MB was 417 mg/g which was greater than that of active carbon (*Marungrueng & Pavasant, 2007*). Some parameters, including the initial dye concentration, pH, temperature, salinity, alga size and dosage, have important effects on the sorption process. In concrete, the adsorption rate constants increased with a decrease of the initial dye concentration. At low dye concentrations (20–80 mg/L), the application of an increasing amount of the alga resulted in a higher percentage of the removed dye(more than 95%) but a lower amount of the dye adsorbed per unit mass (*Marungrueng & Pavasant, 2006*). For MB adsorption, pH of 7–11 might be appropriate because this pH range can supply advantageous surface binding sites of the alga for the ionization of the dye molecule (*Ncibi, Mahjoub & Seffen, 2007*). *Marungrueng & Pavasant (2006)* reported that high temperatures, such as 70 °C, could reduce the adsorption of FGRL, while the maximum adsorption capacity was obtained at 50 °C ($q_m$ for langmuir was 49.26 mg g$^{-1}$). In terms of alga size, a small size of 0.1–0.84 mm resulted in the highest adsorption capacity, followed by intermediate (0.84–2.0 mm) and larger sizes (larger than 2.0 mm) because the small size provided the most surface area and total pore volume for the adsorption of the dye. Additionally, salinity was another stress factor in the system, and high salinity caused a decrease in adsorption capacity due to the competition between Na$^+$ and the dye cations for the binding sites on the algal surface and electrical repulsion (*Punjongharn, Meevasana & Pavasant, 2008*). Furthermore, pseudo second-order kinetic model and Langmuir model could describe the kinetic adsorption and adsorption isotherms process well, respectively (*Punjongharn, Meevasana & Pavasant, 2008*; *Marungrueng & Pavasant, 2006*; *Cengiz & Cavas, 2008*). The sorption process is controlled by both film and pore diffusion (*Marungrueng & Pavasant, 2007*).

As a method of wastewater treatment, dried *C. lentillifera* can adsorb heavy metals and dyes, and fresh *C. lentillifera* can be used as a biofilter in aquaculture systems because it has a significant capacity for nutrient absorption, especially that of NO$_3$-N (*Paul & De Nys, 2008*; *Liu et al., 2016*). *C. lentillifera* was successfully applied at a hatchery scale to a recycling aquaculture system for juvenile spotted babylons (*Babylonia areolata*), and the results revealed that it had a positive effect on the survival rate of spotted babylons, seawater quality and the biomass of *C. lentillifera* (*Chaitanawisuti, Santhaweesuk & Kritsanapuntu, 2011*). In addition, it has often been cultured in shrimp ponds used as water treatment methods (*Chokwiwattanawanit, 2000*).

Besides the application in wastewater treatment, like other algae, bioactive components of *C. lentillifera* and their bioactive potentials have also been studied in recent years.

## Bioactive components of *C. lentillifera* and their biological potentials

*C. lentillifera* contains abundant proteins (10.41% DW (dry weight)), PUFAs (polyunsaturated fatty acids, 16.76% total fatty acids), and total dietary fiber (32.99%

**Table 4   Studies on bioactive components of *C. lentillifera*.**

| Components | Biological activity | References |
|---|---|---|
| Phenolic compounds | Radical-scavenging activity and reducing power ability; Stimulated insulin secretion in pancreatic $\beta$-cells and enhanced glucose uptake | *Matanjun et al. (2008)*, *Nguyen, Ueng & Tsai (2011)*, *Sharma & Rhyu (2014)*, *Sharma, Kim & Rhyu (2017)*, *Sharma, Kim & Rhyu (2015)*, *Abouzid et al. (2014)* |
| Polysaccharides | Increase the phosphorylation of p38 MAPK; Inhibit the proliferation of MCF-7 | *Maeda et al. (2012a)*; *Maeda et al. (2012b)* |
| Siphonaxanthin | cancer-preventing action; Inhibit adipogenesis; | *Ganesan et al. (2011)*; *Li et al. (2015)*; *Zheng et al. (2018)* |

DW) (*Matanjun et al., 2009*; *Nagappan & Vairappan, 2014*), and the alga is also rich in some bioactive components (shown in Table 4).

The total contents of phenolic compounds of dried *C. lentillifera* differed due to the climate and environment in which the alga grew (*Ito & Hori, 1989*). *Nguyen, Ueng & Tsai (2011)* reported that the total phenolic content of thermally dried and freeze-dried *C. lentillifera* were 1.30 mg and 2.04 mg gallic acid equivalent (GAE)/g of dry weight, respectively, which were significantly lower than the data reported by Matajun (30.86% of dry weight; *Matanjun et al., 2008*). As reported in the literature, the phenolic compounds of *C. lentillifera* are often extracted using ethanol, methanol or diethyl ether and show different biological activities. The methanolic and diethyl ether extracts showed better radical-scavenging activity (2.16 mM/mg dry extract by TEAC method) and reducing power ability (362.11 uM/mg dry extract by FRAP method) than those in other brown and red seaweeds (1.63 mM/mg dry extract by TEAC method and 225.00 uM/mg dry extract by FRAP method for *Eucheuma cottonii*; 1.66 mM/mg dry extract by TEAC method and 268.86 uM/mg dry extract by FRAP method) (*Matanjun et al., 2008*). The ethanol extracts had strong hydrogen peroxide-scavenging activity (94.81% with 60 ppm) and weak DPPH-scavenging (IC$_{50}$ was greater than 100 ppm), weak ferric ion-reducing activity (1.93–1.94 ug ascorbic acid equivalent/ml for 20 ppm extract) and weak FIC activity (not exceeding 70% with100 ppm) (*Nguyen, Ueng & Tsai, 2011*). In addition, the ethanol extracts also stimulated insulin secretion in pancreatic β-cells and enhanced glucose uptake by decreasing dipeptidyl peptidase-IV, α-glucosidase and protein-tyrosine phosphatase 1B activities using RIN and 3T3-L1 cells as models (*Sharma & Rhyu, 2014*; *Sharma, Kim & Rhyu, 2017*) and regulated glucose metabolism via the PI3K/AKT signaling pathway in myocytes using L6 cells (*Sharma, Kim & Rhyu, 2015*; *Abouzid et al., 2014*), which could ameliorate insulin resistance.

Polysaccharides are important components of *C. lentillifera* due to their broad spectrum of biological activity. The crude extract of *C. lentillifera* showed anticoagulant property using albino rabbits and the blood of adult dogs. And it exhibited approximate effect of aspirin (*Arenajo et al., 2017*). *Shevchenko et al. (2009)* extracted three polysaccharide fractions, water-soluble P1, P2 and base-soluble P3. The molecular weights of these polysaccharides were 20–60 KDa, 20–40 KDa and more than 70 KDa, respectively. All of the monosaccharide components of these three factions included glucose (Glc), galactose (Gal), mannose (Man) and xylose (Xyl); among these components, glucose was the majority monosaccharide. Moreover, IR spectra of the polysaccharides indicated that

**Figure 2** Structure of siphoxanthin.

the three fractions lacked sulfated groups. However, these results were not inconsistent with those from another study of *Maeda et al. (2012a)*, which reported that the purified polysaccharides (SP1) contained sulfated xylogalactan with a molecular mass >100 KDa. This xylogalactan is mainly composed of galactose, xylose and small quantities of glucose and uronic acid, with 44% sulfation. Furthermore, the SP1 could enhance NO production and activate macrophage cells via NF-κB and increase the phosphorylation of p38 MAPK, which indicates that they can activate RAW 264.7 cells. In another report, β-1,3-xylooligosaccharides could inhibit the proliferation of MCF-7 human breast cancer cells and induce the condensation of chromatin, the degradation of PARP, and the activation of caspase-3/7, which indicates that oligosaccharides can induce apoptosis in MCF-7 cells (*Maeda et al., 2012b*).

Recently, valuable pigments are attracting increasing attention because of their important biological activity. Worth mentioning is siphonaxanthin, a novel and oxidative metabolite of lutein, which is found in *C. lentillifera*. As shown in Fig. 2, its structure contains a conjugated system of 8 C=C double bonds and 1 keto group located at C-8, similar to fucoxanthin. In addition, at the C-19 position, siphonaxanthin has an extra hydroxyl group, which might make it more beneficial than other carotenoids (*Ganesan et al., 2011*; *Walton, Britton & Goodwin, 1973*).

Siphonaxanthin is a specific keto-carotenoid that mainly exists in green algae, such as *Codium fragile*, *C. lentillifera*, *Umbraulva japonica*, and *Caulerpa racemosa*. The content of siphonaxanthin is approximately 0.03%–0.1% of its dry weight (*Sugawara et al., 2014*). Initially, this keto-carotenoid was proved to facilitate the highly efficient energy transfer of carotenoids to chlorophylls (*Akimoto et al., 2008*). Moreover, it might have a largely light-harvesting function in the green light-rich underwater habitat to reduce light damage (*Wang et al., 2013*). In addition to its physiological functions, siphonaxanthin has been found to show many biological activities. It was involved in cancer-preventing action in human leukemia HL-60 cells by increasing in TUNEL-positive cells and chromatin condensation in the cells by decreasing the expression of Bcl-2 but up-regulating the expression of DR5. Furthermore, the anticancer activity of siphonaxanthin was stronger than that of fucoxanthin and siphonein which is an esterified form of siphonaxanthin (*Ganesan et al., 2011*). In addition, siphonaxanthin can show antiobesity effect by inhibiting adipogenesis

in 3T3-L1 preadipocytes and lipid accumulation in the white adipose tissue of KK-Ay mice and inhibiting protein kinase B phosphorylation and regulating the expression of *CEBPA* (enhancer binding protein α), *PPARG* (peroxisome proliferator activated receptor γ), *FABP4* (fatty acid binding protein 4) and *SCD1* (stearoyl coenzyme A desaturase 1) (*Li et al., 2015*). *Zheng et al. (2018)* found that siphonaxanthin can inhibit lipogenesis in hepatocytes by suppressing the excess accumulation of triacylglycerols induced by liver X receptor α agonist and down-regulating nuclear transcription factors with HepG2 cell line.

## SUBHEADINGS

Salinity, nutrients concentration, irradiance and temperature were the most important factors to influence *Caulerpa lentillifera* growth.

Dried seaweed could be used as biosorbent for heavy metals and cationic dyes, and fresh seaweed could be biofilter for the aquaculture system.

The phenolic compounds showed good antioxidant activity and could regulate glucose metabolism.

Polysaccharides and oligosaccharides exhibited immunodulatory effects and cancer-preventing activity.

Siphonaxanthin as a novel function compound showed cancer-preventing activity and lipogenesis inhibiting effect.

In conclusion, *C. lentillifera* need be further studied for more functions such as antiviral, anti-inflammatory areas.

## CONCLUSION

The green seaweed *C. lentillifera* is quite common and popular in Southeast Asian countries and Japan due to its delicious taste and abundant nutrients. During the past 30 years, it has been mass cultivated in some Asian countries, such as the Philippines and Malaysia. And some cultivation conditions, such as the nutrient concentration, salinity, irradiance and temperature, have been studied in relation to the growth of *C. lentillifera*. In addition, this species has been applied to treat wastewater using heavy metal, cationic dye biosorption and aquaculture system. Recently, some bioactive components, such as phenolic compounds, polysaccharides, and siphonaxanthin, have been extracted from *C. lentillifera*, and their biological potentials have also been analyzed by cells. In conclusion, these compounds showed high antioxidant, anticoagulant and immunostimulatory, hypoglycemic, cancer-prevention and lipogenesis inhibition activities, etc. in vitro. It is believed that this seaweed will be a new source of health products with its cultivation at an increasing scale. In addition, perhaps *C. lentillifera* will be used as the resource of biofuel or $CO_2$ fixation just like other algae with further research.

## Funding

This work was supported by the Commonweal Item of the State Oceanic Administration of the People's Republic of China (201505033), NSFC-Shandong joint Fund (U1606403), Shandong Province Key Research and Development Project (2016YYSP010) and Qingdao People's Livelihood Science and Technology Projects (16-6-2-41-nsh). The funders had no role in study design, data collection and analysis, decision to publish, or preparation of the manuscript.

## Grant Disclosures

The following grant information was disclosed by the authors:
Commonweal Item of the State Oceanic Administration of the People's Republic of China: 201505033.
NSFC-Shandong joint Fund:  U1606403.
Shandong Province Key Research and Development Project:  2016YYSP010.
Qingdao People's Livelihood Science and Technology Projects: 16-6-2-41-nsh.

## Competing Interests

The authors declare there are no competing interests.

## Author Contributions

- Xiaolin Chen conceived and designed the experiments, authored or reviewed drafts of the paper, approved the final draft.
- Yuhao Sun performed the experiments.
- Hong Liu contributed reagents/materials/analysis tools.
- Song Liu analyzed the data.
- Yukun Qin prepared figures and/or tables.
- Pengcheng Li check the manuscript.

## Data Availability

The research in this article did not generate any data or code; this is a literature review.

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
