# Peer review of "Advances in cultivation, wastewater treatment application, bioactive components of Caulerpa lentillifera and their biotechnological applications"

_PeerJ, doi:10.7717/peerj.6118_

## Round 0.1 · original submission · Major Revisions

Please address the reviewer comments.

In addition, it has come to our attention that Figure 2 has been copied without attribution from Paul, N. A., Neveux, N., Magnusson, M., & De Nys, R. (2014). Comparative production and nutritional value of “sea grapes”—the tropical green seaweeds Caulerpa lentillifera and C. racemosa. Journal of applied phycology, 26(4), 1833-1844.

Unattributed reproduction of published material without permission is a serious ethical breach. Please provide full attributions and proof that any required permissions have been obtained to reproduce all of your figures under the CC-BY 4.0 license used by PeerJ.

Reviewer 1 ·

Basic reporting

The pictures in Figure 2 are low in resolution and are hard to see clearly. It would be better if pictures with higher qualities can be provided.

In the last senctence of the abstract, it is recommended to replace "various aspects" with the specific points that have been addressed in the paper.

Experimental design

The review is composed of several sections. In each section, one aspect of Caulerpa lentillifera is adequately discussed. One apparent deficiency is that the author did not include smooth transitions between different topics and show his or her understanding of the correlation between these different aspects. The review reads more like a list of facts, rather than a coherently organized paper. It is highly recommended that the author add more transitions between the facts to show the readers why and how the field is developed graduallly.

Validity of the findings

no comment

Reviewer 2 ·

Basic reporting

n/a

Experimental design

n/a

Validity of the findings

n/a

Additional comments

n/a

·

Basic reporting

1- Literature was only focused on Caulerpa lentillifera and its future trends. Only one article was mentioned as a study of C.lentillifera, but it was a study of another alga.
Line 114 This study described the production of rhizoids in Caulerpa prolifera, not of C.lentillifera.

2- Authors show a photo of the alga, that is adequate. However, they also considered a picture of another study and a chemical structure of a compound mentioned. They do not specify if it is a specific chemical structure detected in C.lentillifera. Moreover, tables showing studies with specific data and the most important conclusions are necessary to facilitate the comprehension of the abilities and future trends of C.lentillifera. Tables should emphasize the characteristics of C.lentillifera and its use in different bioprocesses, such as operational conditions and their effect in growth alga,

3- The introduction is clear and concise, but it is necessary to detail the importance of the topics discussed in this review.
Line 62-63 It is necessary to emphasize the importance of the optimum operational conditions (or cultivation conditions) to improve the productivity of the mass culturing of C.lentillifera to supply the current demands.
Line 64-70 I suggest the authors should write one paragraph describing the applications in the industry of this alga. Currently, they are focused in the treatment of wastewaters (specific compounds and the use in aquaculture) and the use of bioactive components.
4- In general, the manuscript is written in professional English, excepting for some sentences. I suggest an English editing by native speakers.

Experimental design

1- Article content is adequate to this journal; however, it is necessary to describe in more detail some studies mentioned. Authors discuss different data of several studies; however, those differences can be for the operational conditions used in them. In addition, the origin of the alga should be considered to discuss the differences, because they could have a different genetical background that can influence the results.

Line 23 This alga also presents saturated fatty acids. It has been described that C.lentillifera contains a high level of unsaturated fatty acids, I suggest to specify this information in the abstract.
Line 26 These are operational conditions (or cultivation conditions) that can produce a stress in specific situations and have been studied in C. lentillifera. In addition, there are other compounds that can produce stress, such as heavy metals.
Line 27 The name species must be written with two words, genera and species, in the first time that is mentioned. After that, the name species can be mentioned as C.lentillifera, in the case of being mentioned again in this review.
Line 81 To understand the contribution of Paul et al., 2014, would be necessary to add more details of the study. Special emphasis on the operational conditions and the origin of alga that possess direct influence in the mass productivity. It is necessary to explain the meaning of "cultivation periods".
Line 82 These are operational conditions or cultivation conditions, characteristics of the environmental. They are stress factors when their values change and produce problems in the physiology of the alga.
Line 89 It is necessary to consider the cultivation conditions in the studies mentioned, as well as the origin of the alga. It is also important to emphasize if seawater or synthetic seawater were used in these studies.
Line 96 It is necessary to know the cultivation conditions that could influence in the differences in specific growth rate in the studies mentioned. These differences can also be related to the origin of the alga
Line 97 Nitrogen possesses different oxidation states more than types of N.
Line 100 To compare the values of the salt concentrations and the growth rate of alga will give more details about their influence on the growth.
Line 102 To compare different ratios is necessary to detail cultivation conditions.
Line 103- 110 It is not clear if phytohormones were incorporated into the assays, or they were hormones of the alga.
Line 111 I suppose the authors wanted to write "cellular enzymes".
Line 117 To describe the cultivation conditions. Guo et al. (2015) performed assays at laboratory scale. They also discuss the effect of different irradiances and other physiological data that could be included in this review.
Line 118- 125 Wu et al. (2017) also discuss other light conditions that could be mentioned as stress factors, as well as the physiological response of the alga.
Line 134 To emphasize the removal efficiency, the values and the conditions of the experiment would support the ability of the dried alga to remove metals.
Line 135 To emphasize the sorption time of the metals, to compare with other studies would support the abilities of this alga.
Line 142 As previously mentioned, to emphasize different aspects of the process, it is necessary to compare with the results of other studies.
Line 146 To know the kinetic model is adequate to understand the process. However, to include the equations are part of other topics and they are not specific of C. lentifillera.
Line 164- 165 To include information of the scaling up of the system would be interesting of considering, since the topic is the use of C.lentillifera in the wastewater treatment.
Line 167- 168 This is a characteristic of the algae, not only of C.lentillifera.
Line 176 To know the removal is better to understand the process, as well as the operational conditions.
Line 180 It is not clear if the effect is in all dyes or in specific cases.
Line 181 To know values would improve the sentence.
Line 185 Salinity is an operational condition mentioned previously, to mention the values detected is better to understand the effect in the sorption capacity.
Line 200 In the wastewater treatment the operational conditions of the process and the removal efficiency are important results that the authors should add in the discussion.
Line 215 To compare values would add information to emphasize the ability of C.lentillifera, a well as details of the assay.
Line 217 It is necessary to know the model used in this study.
Line 220 To specify the experimental conditions and the model used in this study.
Line 232 To consider the differences in the origin of the alga and the cultivation conditions of both studies could give information about the differences in the polysaccharides.
Line 250 These values were described in Codium fragile, Caulerpa lentifillera and Umbraulva japonica. It is not a number detected in several algae.
Line 267 They are operational or cultivation conditions.
Line 271- 272 To mention if assays were in cell culture or animals.
Line 284 To incorporate its use in aquaculture.
Line 287- 288 To mention if they were cell culture or animals.

Validity of the findings

1- Conclusions are according to the information discussed. The information requested would lead to robust conclusions.
2- Argument set out in the introduction was developed. As mentioned above, more details of the studies are requested.
3- The anticoagulant property of crude extract (PMC5604272), the effect of the desiccation (DOI: 10.1007/s10811-018-1442-1) and the
immunostimulatory activity of novel polysaccharides (DOI:10.1016/j.ijbiomac.2017.12.016) are aspects of the C.lentillifera were not mentioned in this review and they have been described in the literature.
4- Conclusions should consider the use at industry scale of the wastewater and animal test of the compounds, as future trends of the use of C.lentillifera.

Additional comments

1-The title of the review should indicate the general applications of C.lentillifera, such as "Advances in cultivation and future trends of C. lentillifera in industrial applications (or biotechnological applications)".
2-Algae have been used in other applications, such as the production of biofuel or their use in CO2 fixation that have not been exploited in C. lentillifera and could be incorporated in the future trends of this alga.

---

## Round 0.2 · Minor Revisions

The manuscript needs further revision.

Reviewer 1 ·

Basic reporting

no comment

Experimental design

no comment

Validity of the findings

no comment

·

Basic reporting

The manuscript was improved, however, I have other comments that could help in the discussion. Authors should revise the measurement units. For example; in line 107 the wrote 0.6 mmol /L, and in line 136 they wrote 40 µmol photons m -2 s-1. They should write the unit according to PeerJ format. It is necessary to consider a space between number and measurement units, for example, line 111 light of 40umol.

Line 37-39 The topics; cultivation, wastewater treatment, and biological active components were previously mentioned in line 24-26. I suggest to delete them in the lines 37-39 and to use the sentence as a short conclusion.
Line 67 It is not clear if this aspect is concerned of the alga or general studies. I suggest writing the sentence again.
Line 91 I suggest changing the word "changed" to the word "change".
Line 94-98 Authors incorporated the origin of the alga, but the sentences are not clear. I write these lines again, taking care of the correct word order in the sentences.
Line 100 What groups?
Line 104 It should be written a conclusion; therefore, studies suggest that the optimal salinity concentration is between...
Line 109 Authors mention the origin of the alga, however, I suggest revising the correct word order.
Line 121 It would be adequate to write a conclusion, such as to mention the best concentration of the nutrients mentioned.
Line 124 Result of what? Results of Tao et al., 2017? It is necessary to detail it.
Line 129 I would delete the word "obviously". It is not obvious the increase of the intracellular crude polysaccharide content, this must be detected.
Line 134 Productivity of biomass?
Authors used the word "decreased" twice in the same sentence, change for a synonym o re-write the sentence.
Line 138 The name of the gene should be in cursive.
Line 138- 139 Authors used the word "different" twice in the same sentence, change for a synonym o re-write the sentence
Line 149 This paragraph should contain some conclusions of the results discussed previously and to include future trends in the cultivation of the alga.
Line 172 It is necessary to write the reference where authors read the information.
Line 215 To write values would help to comprehend the importance of this aspect.
Line 245 I suggest writing: "as a method of wastewater methods".
Line 268 Eucheuma cottonii should be written in cursive.
Line 270- 271 the meaning of the adjectives "strong" and "weak" for the noun “scavenging activities” could be explained in a better way, for example, writing values or other studies to support the information.
Line 289 These results were inconsistent or these results were not consistent.
Line 332 Caulerpa lentillifera should be written in cursive.

Experimental design

The manuscript is adequate. I have some suggestions.
Line 92. To describe some effects in the physiology of the alga would be adequate.
Line 95 It is not clear if the analysis were done at the laboratory or industrial scale.
Line 105-106 All nutrients are important. It is better to write that they are necessary for the growing of the alga and must be taken from the environmental.
Line 165 To mention other materials and the time that they need to take heavy metals will improve the discussion.
Line 325 Information about the role of these genes could help to understand the importance of siphonaxanthin.
Line 328 It is not clear if the inhibition was in cell culture or animals.

Validity of the findings

Line 335- 337 To conclude this information, authors should include information on the topics that have not been studied in C.lentillifera so far

Additional comments

The first part of the review should be revised in detail, there are some paragraphs where the information should be described clearly. The second part, from "Wastewater treatment by C. lentillifera", requires some minor modification.

---

## Round 0.3 · Minor Revisions

The current manuscript is scientifically acceptable.

#

---

## Round 0.4 · accepted · Accept

The current version is acceptable.

#